# The Clinical Impact of Hepatic Arterial Infusion Chemotherapy New-FP for Hepatocellular Carcinoma with Preserved Liver Function

**DOI:** 10.3390/cancers14194873

**Published:** 2022-10-05

**Authors:** Hideki Iwamoto, Takashi Niizeki, Hiroaki Nagamatsu, Kazuomi Ueshima, Joji Tani, Teiji Kuzuya, Kazuhiro Kasai, Youhei Kooka, Atsushi Hiraoka, Rie Sugimoto, Takehiro Yonezawa, Satoshi Tanaka, Akihiro Deguchi, Shigeo Shimose, Tomotake Shirono, Miwa Sakai, Hiroyuki Suzuki, Etsuko Moriyama, Hironori Koga, Takuji Torimura, Takumi Kawaguchi

**Affiliations:** 1Division of Gastroenterology, Department of Medicine, Kurume University School of Medicine, Kurume 830-0011, Japan; 2Iwamoto Internal Medicine Clinic, Kitakyusyu 802-0832, Japan; 3Department of Gastroenterology, Juntendo University, Bunkyo-ku 113-8421, Japan; 4Department of Gastroenterology and Hepatology, Faculty of Medicine, Kindai University, Osaka 589-8511, Japan; 5Department of Gastroenterology and Neurology, Faculty of Medicine, Kagawa University, Miki 761-0793, Japan; 6Department of Gastroenterology and Hepatology, Fujita Healthy University, Toyoake 470-1101, Japan; 7Division of Gastroenterology, IMS Sapporo Digestive Disease Center General Hospital, Sapporo 063-0842, Japan; 8Division of Hepatology, Department of Internal Medicine, Iwate Medical University School of Medicine, Iwate 028-3695, Japan; 9Gastroenterology Center, Ehime Prefectural Central Hospital, Matsuyama 790-0024, Japan; 10Department of Hepato-Biliary-Pancreatology, National Hospital Organization Kyushu Cancer Center, Fukuoka 811-1395, Japan; 11Department of Gastroenterology, Hachinohe Red Cross Hospital, Aomori 039-1104, Japan; 12Department of Gastroenterology and Hepatology, National Hospital Organization Osaka National Hospital, Osaka 540-0006, Japan; 13Department of Gastroenterology, Kagawa Rosai Hospital, Marugame 763-8502, Japan; 14Department of Gastroenterology, Omuta City Hospital, Omuta 836-0861, Japan

**Keywords:** hepatic arterial infusion chemotherapy, sorafenib, propensity score matching, retrospective cohort study, systemic treatment

## Abstract

**Simple Summary:**

New-FP is a regimen of hepatic arterial infusion chemotherapy for hepatocellular carcinoma (HCC). In the study, the regimen significantly prolonged the survival of patients with major portal vein tumor thrombus-HCC.

**Abstract:**

Background: Systemic treatments are recommended for advanced hepatocellular carcinoma (HCC) in preserved liver function. However, their effects are unsatisfactory in some tumor conditions, particularly macrovascular invasion (MVI) including major portal vein tumor thrombus (PVTT). We compared the efficacy of hepatic arterial infusion chemotherapy (HAIC) regimens New-FP and sorafenib for various tumor conditions in preserved liver function. Methods: We retrospectively collected the data of 1709 patients with HCC who were treated with New-FP or sorafenib. Survival was assessed after propensity score matching. Subgroup analyses were conducted: cohort 1 (no MVI or extrahepatic spread (EHS)), cohort 2 (MVI only), cohort 3 (EHS only), cohort 4 (MVI and EHS), and cohort 5 (major PVTT). Results: The New-FP group had a longer median survival time (MST) than the sorafenib in the whole analysis (18 vs. 9 months; *p* < 0.0001). New-FP demonstrated a longer MST compared with sorafenib in cohort 2 and cohort 4. In cohort 5, the MST of the New-FP group was 16 months, while that of sorafenib was 6 months (*p* < 0.0001). For major PVTT-HCC, the response rate of New-FP was 73.0%. The MST of patients who achieved complete response with New-FP was 59 months. Conclusions: HAIC using New-FP is promising for patients with MVI- and major PVTT-HCC in preserved liver function.

## 1. Introduction

Nowadays, systemic treatment has been remarkably developed for hepatocellular carcinoma (HCC) [1]. Sorafenib was the first molecular-targeted agent (MTA) approved for HCC treatment [2]. Currently, atezolizumab plus bevacizumab has become the 1st-line drug [3]. However, sorafenib is still an alternative choice for 1st-line therapy [3]. Systemic treatment is recommended for intermediate-stage HCC refractory to transcatheter arterial chemoembolization and for advanced-stage HCC [4]. 

Macrovascular invasion (MVI) and extrahepatic spread (EHS) are two factors that define advanced-stage HCC [5]. HCC with MVI (MVI-HCC), and especially HCC with portal vein tumor thrombus (PVTT), is a critical condition that can directly cause deterioration of liver function. Although systemic treatment is recommended for advanced-stage HCC, therapeutic effects are not yet satisfactory for MVI-HCC [6]. Finding the best therapeutic strategy represents an unmet medical need.

Locoregional treatments have traditionally been administered in the treatment of HCC. Hepatic arterial infusion chemotherapy (HAIC) is a frequently administered treatment for advanced HCC [7,8]. Recently, the guideline regarding HAIC was also reported [9]. HAIC directly and consecutively delivers chemotherapeutic drugs to the tumor using catheter techniques. According to previous reports, stronger local control (with less toxicity) is an outstanding feature of HAIC as compared with systemic chemotherapy [10]. However, HAIC is not a standard treatment recommended within current guidelines due to the relative sparsity of gold-standard clinical trials conducted to date [11]. Recent clinical trials have provided some evidence supporting the efficacy of HAIC in the treatment of HCC, and the clinical evidence in regard to HAIC has gradually been increasing [12,13]. Herein, we report on the efficacy of a HAIC regimen, “New-FP” (i.e., fine-powder cisplatin (DDP-H, Nippon Kayaku Co. Ltd., Tokyo, Japan) suspended in lipiodol and 5-fluorouracil), in the treatment of advanced HCC. Recently, we conducted a large multicenter retrospective cohort study comparing the efficacy of New-FP and sorafenib to enhance evidence regarding the reproducibility and universality of New-FP [14]. This study revealed the usefulness of locoregional treatment using New-FP. However, questions remain regarding the impact of liver function and tumor condition. We noted that one-third of the analyzed population included patients with Child-Pugh Class B liver function. As the currently approved systemic treatments target patients with preserved liver function, it is necessary to compare the efficacy of New-FP and sorafenib in patients with preserved liver function in order to determine the precise efficacy of this treatment modality. 

Major PVTT, which involves tumor invasion into the 1st branch and trunk of the portal vein, is a life-threatening tumor condition with a short survival prognosis [15]. Although systemic treatments are recommended for major PVTT-HCC, their therapeutic effects are unsatisfactory [16]. Therefore, there is a need to clarify the efficacy of locoregional treatment for major PVTT-HCC. 

This study aimed to clarify the efficacy of New-FP in the treatment of HCC in patients with preserved liver function by comparing the outcomes of New-FP and sorafenib using propensity score matching (PSM) analysis. Additionally, we examined the detailed therapeutic effects of New-FP in patients with major PVTT-HCC. The goal of this study was to support the identification of the best therapeutic strategy for MVI-HCC and major PVTT-HCC in patients with preserved liver function.

## 2. Materials and Methods

### 2.1. Study Design

A multicenter retrospective cohort study was conducted in the study for patients with HCC. The patients with HCC who were initially treated with New-FP or sorafenib were enrolled. The patients were treated in the Kurume Liver Cancer Study Group and at eight other hospitals between March 2009 and June 2019. HCC was diagnosed by biopsy or radiological evaluation. The enrolled patients were >18 years. Death or end-of-study censoring (June 30, 2019) was the complete follow-up period. The patients who could not collect the data were excluded. The Ethics Committee of the Kurume University School of Medicine approved the study (No. 19004). The written informed consent was waived owing to the retrospective study design. 

### 2.2. Patients 

Information of the enrolled patients was collected from patients’ medical records. The clinical stage was judged by the Barcelona Clinic Liver Cancer (BCLC) staging system [5]. HAIC was mainly administered for patients with up to 7 out in BCLC-B or PVTT in BCLC-C. In cases with extrahepatic metastasis, if hepatic lesions were progressed, we considered administration of HAIC to control hepatic lesions first. MVI was diagnosed by enhanced computed tomography or magnetic resonance imaging. Major PVTT includes tumor invasion into the 1st branch (Vp3) and trunk (Vp4) of the poral vein.

### 2.3. Evaluation Items

The following items were evaluated: (i) overall survival (OS), before and after PSM; (ii) factors associated with poor prognosis after PSM; (iii) OS in four cohorts categorized by the presence of MVI and EHS (after PSM), as follows: cohort 1 (no MVI or EHS), cohort 2 (MVI with no EHS), cohort 3 (EHS with no MVI), and cohort 4 (MVI and EHS); and (iv) an OS and forest plot analysis in patients with major PVTT-HCC (regardless of the presence of EHS), before and after PSM (cohort 5).

### 2.4. Propensity Score Matching

Different distributions of covariates between the New-FP and sorafenib groups were adjusted using PSM. The following variables: sex, age, HCC etiology, Child–Pugh class, tumor size, the presence of MVI, the presence of EHS, AFP level, and DCP level were used for PSM. A one-to-one nearest-neighbor matching algorithm with an optimal caliper of 0.3 without replacement was used to generate pairs of patients in each analysis. In each medication group (i.e., both sorafenib and New-FP), there were 198 patients enrolled in the overall analysis, 38 patients in cohort 1, 76 patients in cohort 2, six patients in cohort 3, 38 patients in cohort 4, and 78 patients in cohort 5.

### 2.5. Treatment Protocol

#### 2.5.1. Sorafenib 

Administration of sorafenib (Nexavar; Bayel Co., Ltd., Osaka, Japan) was conducted according to manufacturer recommendations. 

#### 2.5.2. New-FP 

Initially, DDP-H (50 mg/body) suspended in 5–10 mL of lipiodol was administered for patients under angiography, and 5-FU (250 mg bolus injection and 1250 mg continuous injection using an infusion balloon pump) were administered. The dose of the New-FP regimen was determined by the therapeutic response and adverse events. Therapeutic response was assessed using the modified Response Evaluation Criteria in Solid Tumors (mRECIST) criteria [17]. Complete response (CR) is disappearance of intratumoral arterial enhancement. Partial response (PR) is a >30% decrease in viable target lesions. Stable disease (SD) is neither PR nor progressive disease (PD). PD is a >20% increase in viable target lesions.

### 2.6. Statistical Analyses

Data were expressed as counts or means and standard deviations. All statistical analyses were carried out using JMP statistical analysis software (JMP Pro version 14, SAS Institute Inc., Cary, NC, USA). OS was estimated by the Kaplan–Meier method. Factors associated with prognosis were evaluated by univariate and multivariate analysis using Cox proportional hazards models. Variables associated with prognosis (*p* < 0.10) on univariate analysis were entered into the multivariate regression model. Statistical significance was set to a two-tailed *p*-value of <0.05.

## 3. Results

### 3.1. Patient and Tumor Characteristics

The flow diagram for study enrollment is shown in Figure 1. A total of 1709 consecutively presenting patients diagnosed with HCC, including 671 in the New-FP group and 1038 in the sorafenib group, were enrolled in this study. Among them, 436 patients were excluded from the present analysis because of Child–Pugh class B or C liver function categorizations. Additionally, 11 patients were excluded because of incomplete data. Finally, 1262 patients, including 418 in the New-FP group and 844 in the sorafenib group, were enrolled in the PSM analysis.

Patient characteristics are summarized in Table 1. Etiology, tumor size, the presence of MVI and EHS, BCLC stage, and mean DCP level were statistically significantly different between the two groups before PSM. However, statistical differences disappeared after PSM (Table 1).

### 3.2. Overall Survival 

Figure 2A,B shows the survival curves for the New-FP and sorafenib groups before and after PSM. Before PSM, the MST for New-FP and sorafenib was 17 and 12 months, respectively. A statistically significant difference was observed between the groups (Figure 2A, hazard ratio (HR) 0.68, 95% confidence interval (CI): 0.59–0.79); *p* < 0.0001). After PSM, the MSTs for New-FP and sorafenib were 18 and 9 months, respectively, with a statistically significant difference between the groups (Figure 2B, HR 0.54; 95% CI 0.43–0.68, *p* < 0.0001).

### 3.3. Prognostic Factors

After applying PSM, prognostic factor analyses were performed for the entire dataset (Table 2). A Child–Pugh score of 5, the presence of major PVTT, the presence of EHS, the better therapeutic response, and New-FP treatment were associated with patient prognosis on univariate analysis. Multivariate analysis showed that a Child–Pugh score of 5 and New-FP were each independent better prognostic factors. The presence of major PVTT and EHS were independent poor prognostic factors.

### 3.4. Subgroup Analyses

Subgroup analyses were conducted in the aforementioned five cohorts (Figure 2C–F).

#### 3.4.1. Cohort 1: No MVI or EHS

Patient and tumor characteristics for cohort 1 are shown in Appendix A. Before PSM, there were statistically significant differences in sex, etiology, and tumor size between the two groups. After PSM, these statistical differences disappeared. The OS including both groups in cohort 1 was 19 months The MSTs in the New-FP and sorafenib groups were 20 and 17 months, respectively (Figure 2C). There was no statistically significant difference in OS between them (*p* = 0.28, HR 0.71, 95% CI 0.39–1.32).

#### 3.4.2. Cohort 2: MVI without EHS

Patient and tumor characteristics for cohort 2 are shown in Appendix A. Before PSM, there was a statistically significant difference in tumor size between the two groups. After PSM, the statistical difference disappeared. The OS including both groups in cohort 2 was 12 months The MSTs in the New-FP and sorafenib groups were 19 and eight months, respectively (Figure 2D). New-FP statistically significantly prolonged survival compared with sorafenib (*p* < 0.0001, HR 0.51, 95% CI 0.35–0.74).

#### 3.4.3. Cohort 3: EHS without MVI

Patient and tumor characteristics for cohort 3 are shown in Appendix A. Before PSM, there were statistically significant differences in age or tumor size between the two groups. After PSM, the statistical differences disappeared. The OS including both groups in cohort 3 was 11 months The MSTs for New-FP and sorafenib were 13 and 7.5 months, respectively (Figure 2E). There was no statistically significant difference in OS between the two groups (*p* = 0.095, HR 1.34, 95% CI 0.37–4.82).

#### 3.4.4. Cohort 4: MVI and EHS

Patient and tumor characteristics for cohort 4 are shown in Appendix A. Before PSM, there was a statistically significant difference in tumor size between the two groups. After PSM, the statistical difference between the groups disappeared. The OS including both groups in cohort 4 was 7 months. The MSTs of the New-FP and sorafenib groups were 8.8 months and 4 months, respectively (Figure 2F). New-FP statistically significantly prolonged survival as compared with sorafenib (*p* = 0.003, HR 0.50, 95% CI 0.31–0.82).

### 3.5. Assessment in Major PVTT-HCC

#### 3.5.1. Comparison of OS in Major PVTT-HCC 

In cohort 5, the OS of patients with major PVTT-HCC was compared between the New-FP and sorafenib groups. Patient and tumor characteristics for cohort 5 are shown in Appendix A. Before PSM, there were statistically significant differences in tumor size and the presence of EHS and DCP levels between the two groups. After PSM, the statistical differences between the groups disappeared. The MSTs of the New-FP and sorafenib groups were 13 and six months, respectively, before applying PSM (Figure 3A). The MSTs of the New-FP and sorafenib groups were 16 and six months, respectively, after applying PSM (Figure 3B).

#### 3.5.2. Forest Plot Analysis 

We performed a forest plot analysis to shed light on the factors associated with prognosis in patients with major PVTT-HCC (Figure 4). Sorafenib, an AFP level of ≥400 ng/mL, the presence of EHS, the presence of Vp4 (portal vein invasion in the main trunk), tumor size ≥5 cm, and age ≥65 years were poor prognostic factors in patients with major PVTT-HCC.

#### 3.5.3. Efficacy of New-FP in Major PVTT-HCC

Data on the therapeutic response to New-FP for major PVTT-HCC are shown in Figure 3C. The CR, PR, SD, and PD rates were 16.6%, 56.4%, 16.7%, and 10.3%, respectively. The objective response rate was 73.0%, and the disease control rate was 89.7%. The MSTs of responders and non-responders were 23 and six months, respectively (*p* = 0.0043, HR 0.48, 95% CI 0.29–0.79, Figure 3D). The MSTs of patients who achieved CR, PR, SD, and PD were 59, 17.2, 8, and 4 months, respectively (*p* < 0.0001, Figure 3E).

## 4. Discussion

This study demonstrated the efficacy of New-FP under various tumor conditions in patients with preserved liver function. New-FP was shown to be highly effective in patients with MVI regardless of the presence of EHS. In the prognostic analysis of the whole dataset conducted after applying PSM, the factors associated with better prognosis were a Child–Pugh score of 5 and New-FP, and the factors associated with poor prognosis were the presence of major PVTT and EHS. New-FP also showed dramatic efficacy in patients with major PVTT-HCC. The objective response rate to New-FP for major PVTT-HCC was >70%, and the MST for the responders was >20 months. Notably, 16.6% of patients with major PVTT-HCC achieved CR on applying New-FP; the MST of the patients who achieved CR was 59 months. 

This study is the largest retrospective cohort study comparing the efficacy of a single HAIC regimen and systemic treatment for HCC in patients with preserved liver function. In this study, the efficacy of New-FP was drastically enhanced by excluding patients with poor liver function. Ueshima et al. similarly reported a large-scale retrospective cohort study examining the efficacy of HAIC and sorafenib for HCC in patients with preserved liver function [18]. The HAIC regimens evaluated in their study included cisplatin monotherapy, interferon-α + 5-FU, and low-dose FP; the MSTs of patients with HCC with MVI and without EHS were 10.6 months for HAIC and 9.1 months for sorafenib. HAIC was statistically significantly superior to sorafenib alone for MVI-HCC. Although a direct comparison between the referenced report and the current study cannot be made, we note that the MST for New-FP was superior to that of HAIC in this prior study. Recently, our group reported a comparative study between New-FP and a conventional HAIC regimen of low-dose FP [19]; the study revealed that New-FP prolonged patient survival as compared to low-dose FP. These results suggest that applying a HAIC regimen is important in the treatment of HCC. 

In this study, New-FP showed strong efficacy for MVI-HCC, regardless of the presence of EHS. Systemic treatment is recommended for patients with HCC involving EHS [20]. However, the degree of progression of hepatic lesions often determines the prognosis of patients with HCC involving EHS [21]. Our previous report showed that primary tumor stage T4 was an independent prognostic factor in patients with HCC and EHS [22]. Therefore, locoregional treatment is important even for HCC with EHS. To find the appropriate therapeutic strategy for HCC with EHS, we have to properly assess the progression of hepatic lesions. Then, we need to select the best treatment. Further analyses are needed to clarify the clinical question regarding treatment for HCC with EHS.

The prognosis for patients with major PVTT-HCC is extremely poor. For example, the basal prognosis of such patients is three months for Vp4 and six months for Vp3 (right or left portal vein invasion) [23]. The previously reported prognosis of patients with major PVTT-HCC treated with sorafenib was 6.5 months [24]. The prognosis for other 1st-line systemic treatments, such as lenvatinib and atezolizumab plus bevacizumab combination therapy, is also approximately six months [7,25]. In the SILIUS trial, a randomized clinical trial comparing the efficacy of sorafenib and sorafenib plus low-dose FP in the treatment of HCC, additional low-dose FP showed a statistically significant effect for major PVTT-HCC as compared with sorafenib [26]; these researchers also reported that objective response could be an independent prognostic factor for OS [27]. In our current study, responders to New-FP demonstrated longer survival. Surprisingly, the MST of CR cases reached nearly five years. Achieving a therapeutic response is critically important in the treatment of major PVTT-HCC, and evidence indicates that New-FP should be selected for these patients. 

Currently, we are in an era of systemic treatments for HCC, and sequential drug therapy has become mainstream [28]. However, there are many locoregional treatments available for HCC as well. Therefore, we need to find an optimal therapeutic strategy that combines systemic and locoregional treatments. In the study, the multivariate analysis revealed that the poor prognostic factors of the New-FP/HAIC group before PSM were a Child–Pugh score of 6, the presence of major PVTT, and EHS, respectively. To manage the patients with these factors, New-FP monotherapy might be not enough. Ikeda et al. reported on the usefulness of a combination therapy (cisplatin monotherapy and sorafenib) [29]. We also previously reported on the efficacy of combination therapy with TACE or HAIC and lenvatinib [30], and Kudo et al. reported on the importance of conversion treatment after atezolizumab plus bevacizumab combination therapy [31]. These reports suggest that multidisciplinary treatment combining systemic and locoregional treatments, including HAIC, might be the best therapeutic strategy for the management of advanced HCC. From this perspective, data available to date strongly suggest that New-FP, which demonstrates a high response rate, can be an effective choice in the era of multisystemic treatments. 

Although the study revealed the effectiveness of New-FP for PVTT-HCC, the best treatment for BCLC-B still remains unclear. In the study, cohort 1 is the analysis for BCLC-B. There was no significant difference in the MSTs between New-FP and sorafenib. One of the reasons why there is no difference is that tumor conditions are too heterogeneous. TACE is recommended for BCLC-B. However, the systemic treatment is also recommended for TACE unsuitable cases in BCLC-B. So far, there have been no clinical trials which directly compared with TACE, HAIC, and the systemic treatment in the area. What we know in the treatment of BCLC-B is that achieving complete response/cancer-free prolongs survival and sequential therapy, e.g., TACE to systemic treatments, TACE to HAIC, or HAIC to systemic treatments, can prolong the time to progression to BCLC-C [32]. Further analyses are needed to find the best treatment for BCLC-B.

This study has several limitations. First, this study was retrospective in nature. Second, although PSM was conducted, the ratio of tumor conditions (such as in regard to MVI) might be favorable for New-FP; therefore, we performed subgroup analyses. Third, the number of patients in cohort 3 was very low. There is a possibility that selection bias might have affected the outcomes in cohort 3. Additionally, the data regarding post-treatments were not collected because of the issue due to the multicenter study. Influence of post-treatments significantly affects patients’ survival. To overcome these limitations, we need to perform a randomized prospective study to compare New-FP and various MTAs, including sorafenib, in the future. 

## 5. Conclusions

In conclusion, we found that New-FP meaningfully prolonged the OS of patients with HCC with preserved liver function, especially patients with MVI-HCC, regardless of the presence of EHS. Moreover, New-FP showed statistically significant efficacy in patients with major PVTT-HCC. We conclude that New-FP is a promising HAIC regimen that can provide a chance of survival for more than five years even for patients with major PVTT-HCC.

## Figures and Tables

**Figure 1 cancers-14-04873-f001:**
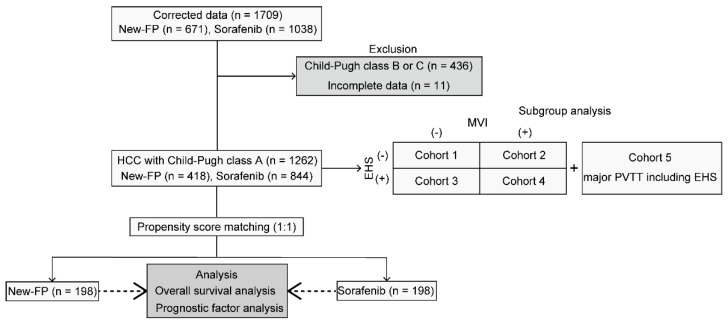
Flowchart illustrating the study inclusion and exclusion process. EHS, extrahepatic spread; MVI, macrovascular invasion; New-FP, fine-powder cisplatin and 5-fluorouracil; PVTT, portal vein tumor thrombus.

**Figure 2 cancers-14-04873-f002:**
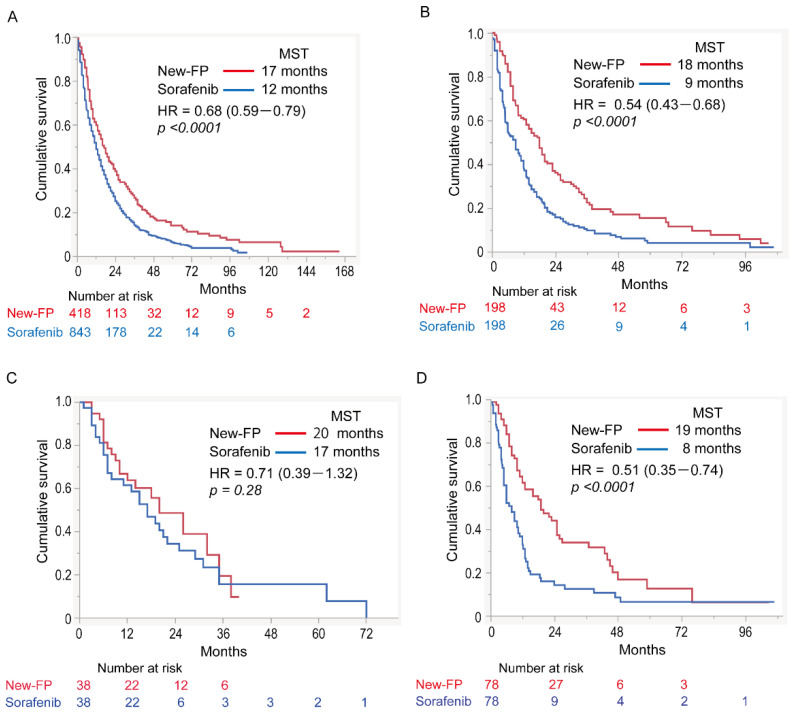
Overall survival curves for patients in the New-FP and sorafenib groups. (**A**) New-FP (*n* = 418, red line) and sorafenib (*n* = 844, blue line), before propensity score matching. (**B**) New-FP (*n* = 198, red line) and sorafenib (*n* = 198, blue line), after propensity score matching. (**C**) Cohort 1 (no MVI or EHS), New-FP (*n* = 38, red line) and sorafenib (*n* = 38, blue line). (**D**) Cohort 2 (MVI with no EHS), New-FP (*n* = 78, red line), and sorafenib (*n* = 78, blue line). (**E**) Cohort 3 (EHS with no MVI), New-FP (*n* = 6, red line), and sorafenib (*n* = 6, blue line). (**F**) Cohort 4 (MVI and EHS), New-FP (*n* = 78, red line), and sorafenib (*n* = 78, blue line). HR, hazard ratio; MST, median survival time; New-FP, fine-powder cisplatin and 5-fluorouracil; EHS, extrahepatic spread; MVI, macrovascular invasion.

**Figure 3 cancers-14-04873-f003:**
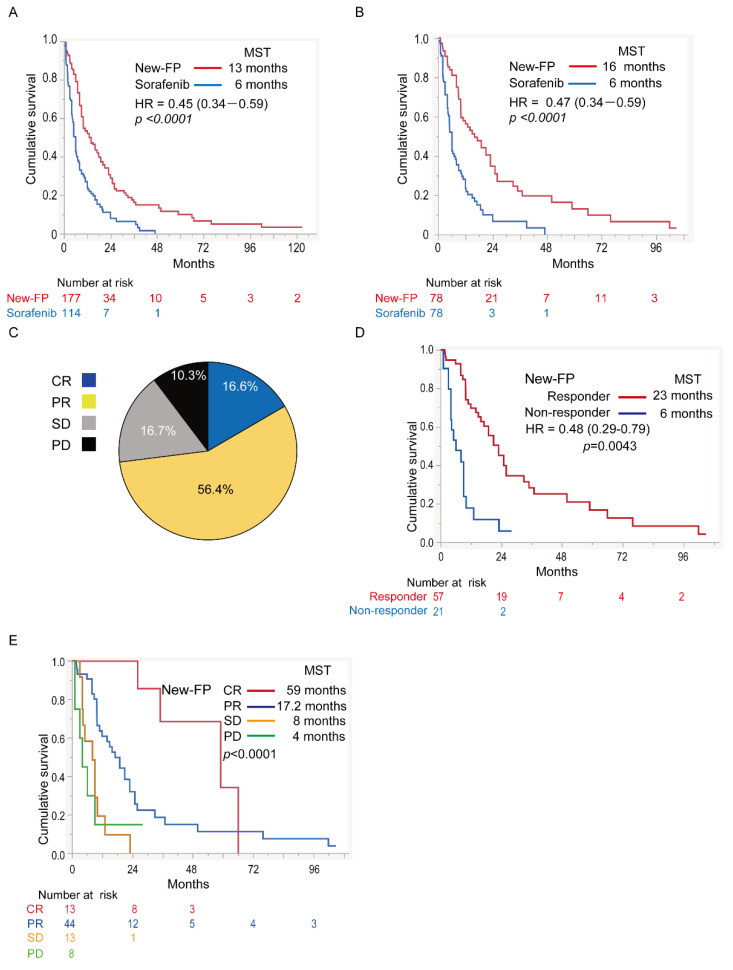
Therapeutic effects of New-FP and sorafenib in HCC with major PVTT. (**A**) Cohort 5, survival curves of patients with major PVTT-HCC before propensity score matching: New-FP (*n* = 177, blue line) and sorafenib (*n* = 114, red line). (**B**) Cohort 5, survival curves of patients with major PVTT-HCC after propensity score matching: New-FP (*n* = 78, blue line) and sorafenib (*n* = 78, red line). (**C**) Therapeutic response for New-FP in major PVTT-HCC. (**D**) Survival curves in the responder (red line, *n* = 57) and non-responder groups (blue line, *n* = 21) for New-FP. (**E**) Survival curves in different therapeutic response groups for New-FP: CR (red line, *n* = 13), PR (blue line, *n* = 44), SD (yellow line, *n* = 13), and PD (green line, *n* = 8). CR, complete response; HCC, hepatocellular carcinoma; HR, hazard ratio; MST, median survival time; PD, progressive disease; PR, partial response; PVTT, portal vein tumor thrombus; SD, stable disease.

**Figure 4 cancers-14-04873-f004:**
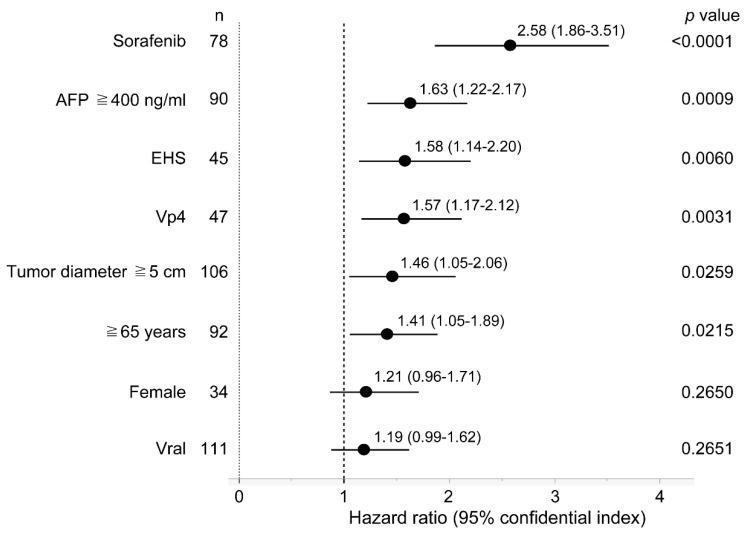
Forest plot analysis for factors associated with poor prognosis in patients with major PVTT-HCC. AFP, alpha-fetoprotein; EHS, extrahepatic spread; HCC, hepatocellular carcinoma; Vp4, tumor invasion into the trunk of the portal vein; PVTT, portal vein tumor thrombus.

**Table 1 cancers-14-04873-t001:** Patient characteristics before and after propensity score matching.

	Before Matching *n* = 1262			After Matching *n* = 396		
Patient Characteristics	New-FP *n* = 418	Sorafenib *n* = 844	*p*-Value	New-FP *n* = 198	Sorafenib *n* = 198	*p*-Value
Age (years)	68.6 ± 10.9	70.1 ± 9.61	0.9398	66.6 ± 10.9	65.3 ± 10.7	0.8046
Sex (male/female)	327/91	677/167	0.4154	62/16	60/18	0.8127
HCV/HBV/non-viral	181/77/160	465/154/225	<0.0001	37/19/22	27/28/23	0.8074
Child–Pugh score (5/6)	251/167	526/317	0.0563			0.1263
Tumor characteristics						
Tumor size (cm)	8.29 ± 4.60	3.59 ± 3.39	<0.0001	7.62 ± 4.21	7.26 ± 4.75	0.8107
MVI (+/−)	357/61	605/239	<0.0001	143/55	146/52	0.7342
Severe PVTT (+/−)	177/180	114/491	<0.0001	117/81	78/120	0.2304
EHS (+/−)	76/342	429/415	<0.0001	24/54	21/57	0.7385
BCLC stage (B/C)	52/366	305/532	<0.0001	46/152	50/148	0.6390
AFP (ng/mL)	27,729.0 ± 154,561.1	17,735.3 ± 110,114.8	0.1886	48,645.45 ± 248,475.48	52,073.81 ± 163,296.82	0.6153
DCP (mAU/mL)	23,513.0 ± 70,955.4	14,373.38 ± 71,899.8	0.0339	30,379.13 ± 95,631.52	27,714.78 ± 61,755.77	0.8773

AFP, alpha-fetoprotein; BCLC, Barcelona Clinic Liver Cancer; DCP, des-gamma carboxyprothrombin; EHS, extrahepatic spread; PVTT, portal vein tumor thrombus; HBV, hepatitis B virus; HCV, hepatitis C virus; MVI, macrovascular invasion.

**Table 2 cancers-14-04873-t002:** Univariate and multivariate analyses for evaluating factors associated with overall survival.

Variable	Univariate Analysis (*p*-Value)	Multivariate Analysis (*p*-Value)	Hazard Ratio (95% CI)
Sex (male/female)	0.1681		
HBV (+/−)	0.3909		
HCV (+/−)	0.1851		
Child–Pugh score (5/6)	0.0002	0.0068	0.72 (0.57–0.91)
BCLC stage (B/C)	0.2621		
MVI (+/−)	0.9679		
Major PVTT (+/−)	0.0213	0.0002	1.57 (1.24–1.99)
EHS (+/−)	<0.0001	0.0011	1.76 (1.37–2.26)
AFP (≥400 ng/mL/<400 ng/mL)	0.1811		
Better therapeutic response	<0.0001	0.7901	
Treatment (New-FP/sorafenib)	0.0350	<0.0001	0.52 (0.41–0.65)

AFP, alpha-fetoprotein; BCLC, Barcelona Clinic Liver Cancer; CI, confidence interval; EHS, extrahepatic spread; HBV, hepatitis B; HBC, hepatitis C; MVI, macrovascular invasion; New-FP, fine-powder cisplatin and 5-fluorouracil.

## Data Availability

Data are contained within the article or Appendix A.

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
