# Peer review of "The Clinical Impact of Hepatic Arterial Infusion Chemotherapy New-FP for Hepatocellular Carcinoma with Preserved Liver Function"

_cancers, 2022, doi:10.3390/cancers14194873_

Round 1

Reviewer 1 Report

1.    The authors reported the therapeutic outcomes a large series of HCC patients during a 10-year period. The data is informative and valuable.

2.    The manuscript is entitled as “… with Major Portal Vein Tumor Thrombus”, but many non-PVI, PV1 or PV2 patients were included. This made heterogeneity of this study.

3.    The HAIC indications in the authors’ institutes should be described in the text.

4.    In cohot-1 patients, the tumor size was not large (<5 cm) with no MVI or EHS. What’s the most ideal
treatment for these BCLC-B patients should be discussed (TACE or HAIC). Although the median OS was good (20 vs 17 months, respectively), IMbrave 150 study reported the median OS of unresectable HCC to be 19.2 months with atezolizumab plus bevacizumab and 13.4 months with sorafenib, and moreover,
Llovet (Lancet 2002) reported 17.9 months in their HCC patients under supportive care (28.7 months with TACE). The above reported OS from different treatment regimens/modalities may confuse readers. Thereafter, a strict stratification of the patients’ status is essential. I suggest delete the cohort-1 and focus on the patients either with MVI or major PVTT.

5.    After PSM, inconsistent MST of sorafenib group for all patients (11 or 9 months?)

6.    The risk factors of HAIC before PSM should be analyzed and reported.

7.    The OS in each cohort should also be reported.

8. Your study showed HCC patients with EHS had poorer MST, but still significantly longer than those of sorafenib group (13 vs 7.5 months in without MVI and 8.8 vs 4 months in with PVI). This data is important as many clinicians considered EHS as a contraindication for locoreginal therapy. This point should be more emphasized.

Author Response

Thank you very much for your comments regarding our manuscript (cancers-1896612). Your comments have helped us to improve our manuscript. In line with your comments, please find below our point-by-point responses.

Reviewer: 1
1.    The authors reported the therapeutic outcomes a large series of HCC patients during a 10-year period. The data is informative and valuable.

Answer

Thank you very much for your comments and for understanding the important points of our manuscript.

2. The manuscript is entitled as “… with Major Portal Vein Tumor Thrombus”, but many non-PVI, PV1 or PV2 patients were included. This made heterogeneity of this study.

Answer

We totally agree with your comment. According to your comment, we have changed the title to “The clinical impact of Hepatic Arterial Infusion Chemotherapy New-FP for Hepatocellular Carcinoma with the preserved liver function ” to include the whole concept of the study.

3. The HAIC indications in the authors’ institutes should be described in the text.

Answer

Thank you for your important suggestion. We administer our HAIC treatment for BCLC-B (Up to 7 out) and BCLC-C. In BCLC-C cases, if hepatic lesions are progressed, we administer our HAIC treatment first even for the cases with extrahepatic spread. We added the description regarding the HAIC indications in Line 114 to 118.

4. In cohort-1 patients, the tumor size was not large (<5 cm) with no MVI or EHS. What’s the most ideal treatment for these BCLC-B patients should be discussed (TACE or HAIC). Although the median OS was good (20 vs 17 months, respectively), IMbrave 150 study reported the median OS of unresectable HCC to be 19.2 months with atezolizumab plus bevacizumab and 13.4 months with sorafenib, and moreover, Llovet (Lancet 2002) reported 17.9 months in their HCC patients under supportive care (28.7 months with TACE). The above reported OS from different treatment regimens/modalities may confuse readers. Thereafter, a strict stratification of the patients’ status is essential. I suggest delete the cohort-1 and focus on the patients either with MVI or major PVTT.

Answer

Thank you very much for your valuable comment and suggestion. I totally agree with you about the difficulty to find the ideal treatments for BCLC-B. I really understand that Cohort 1 might give confusion to the readers. However, I think you can understand that it is also difficult to delete Cohort 1 as the study design in the study. The study wanted to show which tumor conditions are suitable for HAIC New FP and sorafenib. So, we added detailed descriptions regarding the ideal treatments for BCLC-B in the discussion part.

In BCLC-B patients, I think we cannot decide on the ideal treatment because tumor conditions in BCLC-B are too heterogeneous. However, we know achieving complete response/ cancer-free is the strongest factor for prolongation of patients with BCLC-B. And we know the sequential therapy, e.g. TACE to the systemic therapy, TACE to HAIC, HAIC to the systemic therapy, is the ideal therapeutic strategy for BCLC-B, which can prolong the time to progression to BCLC-C. We added the descriptions in the discussion part (line 345 to 355 as new paragraph). Your suggestions improved our manuscript. Thank you very much.

5. After PSM, inconsistent MST of sorafenib group for all patients (11 or 9 months?)

Answer

Thank you for your comment. The MST of sorafenib group was 11 months. The MST of sorafenib described in the Abstract is wrong. we apologize for the mistake. We corrected it.

6. The risk factors of HAIC before PSM should be analyzed and reported.

Answer

We appreciate your suggestion. We have conducted the analysis for the risk factor of HAIC before PSM. The multivariate analysis revealed Child-Pugh score 6, the presence of PVTT, major PVTT, and the presence of extrahepatic metastasis were independent prognostic factors, respectively. These results are also very informative, which helped to improve our manuscript. Thank you very much. We added the description of the analysis in the line 333 to 336. Thank you very much for your valuable comment. 

7. The OS in each cohort should also be reported.

Answer

Thank you for your comment. We have shown the OS in each cohort. The OS in cohort 1 (the data after PSM) was 19 months. The OS in cohort 2 (the data after PSM) was 12 months. The OS in cohort 3 (the data after PSM) was 11 months. The OS in cohort 4 (the data after PSM) was 7 months. We added the descriptions in the results part (lines 215-216, 222 and 223, 229-230, 236-237.)

8. Your study showed HCC patients with EHS had poorer MST, but still significantly longer than those of sorafenib group (13 vs 7.5 months in without MVI and 8.8 vs 4 months in with MVI). This data is important as many clinicians considered EHS as a contraindication for locoreginal therapy. This point should be more emphasized.

Answer

We totally agree with your comments. As you mention, the degree of hepatic lesions is one of the important factors associated with the prognosis of patients with HCC with EHS. Now, systemic treatment is recommended for HCC with EHS. However, local control of hepatic lesions using locoregional treatment including HAIC might prolong the survival of patients with EHS. Further analyses are needed to clarify the clinical question. We added these descriptions in the discussion part (lines 307-315). Thank you very much for your constructive and valuable comments.

Reviewer 2 Report

the article is well written and very important to the field and can be published as it is

Author Response

Thank you very much for your agreement. We really appreciate your understanding of the importance of our manuscript.

Reviewer 3 Report

This article shows that hepatic arterial infusion chemotherapy (HAIC) using New-FP significantly prolonged the survival of hepatocellular carcinoma (HCC) patients with major portal vein tumor thrombus in preserved liver function compared with sorafenib therapy. The article was interesting because of analyzing a large population. However, there are some problems in this article.

Major point:

1.     As shown in Table 2, the authors evaluated prognostic factors associated with overall survival. Therapeutic response and post-therapy after first-line therapy were not included in this analysis. You should add these factors, right?

Minor points:

1.     In the study design, you should describe the inclusion and exclusion criteria in this study.

2.     This is a multi-center cohort study. You should show the approved numbers of other institutions.

3.     There were no data regarding the degree of macrovascular invasion (MVI). There were two types of MVI (portal vein invasion, hepatic vein invasion). In addition, you should describe the definition of “severe PVTT”. Is “Major PVTT” defined as the tumor thrombus in the trunk of portal vein?

4.     In the Fig.4, the authors showed AFP ≥400ng/mL, tumor diameter ≥5 cm, and age ≥65 years. In contrast, you demonstrated AFP >400ng/mL, tumor diameter >5 cm, and age >65 years in the text. There was the difference of these inequality signs between the figure and the text.

Author Response

To reviewer 3,

Thank you very much for your comments regarding our manuscript. Your comments have helped us to improve our manuscript. In line with your comments, please find below our point-by-point responses.

Major point:

1. As shown in Table 2, the authors evaluated prognostic factors associated with overall survival. Therapeutic response and post-therapy after first-line therapy were not included in this analysis. You should add these factors, right?

 Answer

Thank you very much for your comments. We have conducted the analysis of prognostic factors with overall survival including the therapeutic response. With respect to post-therapy after first-line therapy, we do not have the data regarding post-therapy because the data were collected from the multi-facilities, unfortunately. I have changed Table 2. The therapeutic response was a significant factor when conducting the univariate analysis. The multivariate analysis showed the presence of major PVTT, the presence of EHS and Child-Pugh score 6 were independent poor prognostic factors, respectively. We added the description regarding the analysis. Thank you for your important comment. And I’m sorry that I could not answer about post-treatment. We totally agree with the importance of post-treatment. We added the description of post-treatment in the limitation section of the discussion part (lines 360-362).

Minor points:

1. In the study design, you should describe the inclusion and exclusion criteria in this study.

Answer

Thank you for your valuable comment. We described the inclusion and exclusion criteria in lines 103 to 108 and line 114 to 116.

2. This is a multi-center cohort study. You should show the approved numbers of other institutions.

Answer

Thank you very much for your comments. No31-9 in Ehime Prefectural Central Hospital, 2019-0159 in Fujita Health University, 2021-159 in Kagawa University, 19064 in Osaka National Hospital, 2019-13 in National Hospital Organization Kyushu Cancer Center, H18-154 in Juntendo University. In several institutions, there were no appropriate approved ethical numbers for the study. Alternatively, the opt-out agreement regarding the study is presented in our facility's homepage, which describes all facilities in the study (line 388-393).

3. There were no data regarding the degree of macrovascular invasion (MVI). There were two types of MVI (portal vein invasion, hepatic vein invasion). In addition, you should describe the definition of “severe PVTT”. Is “Major PVTT” defined as the tumor thrombus in the trunk of portal vein?

Answer

Thank you for your question. As you mention, there are two types of MVI in HCC. In the study, there were no detailed information about the content of MVI except major PVTT because the data were collected from the multi-centers. The definition of major PVTT is tumor invasion into 1st branch or trunk of the portal vein. We added the descriptions about them in the line 116-118.

4. In the Fig.4, the authors showed AFP ≥400ng/mL, tumor diameter ≥5 cm, and age ≥65 years. In contrast, you demonstrated AFP >400ng/mL, tumor diameter >5 cm, and age >65 years in the text. There was the difference of these inequality signs between the figure and the text.

Answer

Thank you very much for your point out. These are typos. We apologize for these mistakes. The descriptions in figure 4 are correct. We revised them (line 265-267).

Round 2

Reviewer 3 Report

The revised article is almost well-written. However, there was a mistake and a problem in the revised article.

1.     In the abstract section and the text, the MSTs for New-FP and sorafenib were 18 and 11 months, respectively, after PSM. However, in Figure 2 (B), the MSTs for New-FP and sorafenib were 18 and 9 months, respectively.

2.     On page 11, lines 330-332, “In the study, the poor prognostic factors of New FP/HAIC group before PSM were Child-Pugh score 6, the presence of PVTT, major PVTT, and EHS, respectively (data not shown)”. Is this the univariate analysis or multivariate analysis? The presence of PVTT and major PVTT are confounders.

Author Response

Thank you very much for your comments. And we apologize for our careless mistakes.   

  1. In the abstract section and the text, the MSTs for New-FP and sorafenib were 18 and 11 months, respectively, after PSM. However, in Figure 2 (B), the MSTs for New-FP and sorafenib were 18 and 9 months, respectively.

Answer

Thank you very much for your comment. We sincerely apologize for our careless mistake. I revised it. 9 months was correct.

  1. On page 11, lines 330-332, “In the study, the poor prognostic factors of New FP/HAIC group before PSM were Child-Pugh score 6, the presence of PVTT, major PVTT, and EHS, respectively (data not shown)”. Is this the univariate analysis or multivariate analysis? The presence of PVTT and major PVTT are confounders.

Answer

Thank you very much for your point out. This was the result of the multivariate analysis. And the presence of major PVTT was the factor. The presence of PVTT should be removed. We revised them (lines 330-331). We apologize for our mistake.

We appreciate your comment.